# Forced Convective Heat Transfer Coefficient Measurement of Low Concentration Nanorods ZnO–Ethylene Glycol Nanofluids in Laminar Flow

**DOI:** 10.3390/nano12091568

**Published:** 2022-05-05

**Authors:** Md. Shah Alam, Bodrun Nahar, Md. Abdul Gafur, Gimyeong Seong, Muhammad Zamir Hossain

**Affiliations:** 1Department of Chemistry, Jagannath University, Dhaka 1100, Bangladesh; m170301033@chem.jnu.ac.bd (M.S.A.); bodrunnahar@chem.jnu.ac.bd (B.N.); 2Pilot Plant & Process Development Centre, Bangladesh Council of Scientific and Industrial Research (BCSIR), Dhaka 1205, Bangladesh; d_r_magafur@bcsir.gov.bd; 3New Industry Creation Hatchery Center, Tohoku University, 6-6-10 Aoba, Aramaki, Aoba-ku, Sendai 980-8579, Japan

**Keywords:** hydrothermal, zinc oxide nanorods, zinc oxide–ethylene glycol NF, zeta potential, ultraviolet-visible spectroscopy, shell and tube heat exchanger, convective HTC, laminar flow

## Abstract

This paper presents the experimental forced convective heat transfer coefficient (HTC) of nanorods (NRs) zinc oxide–ethylene glycol nanofluids (ZnO–EG NFs) in laminar flow. First, ZnO NRs were synthesized using a hydrothermal method that uses zinc acetate dihydrate [Zn(CH_3_COO)_2_·2H_2_O] as a precursor, sodium hydroxide as a reducing agent, and polyvinylpyrrolidone (PVP) as a surfactant. The hydrothermal reaction was performed at 170 °C for 6 h in a Teflon-lined stainless-steel tube autoclave. The sample’s X-ray diffraction (XRD) pattern confirmed the formation of the hexagonal wurtzite phase of ZnO, and transmission electron microscopy (TEM) analysis revealed the NRs of the products with an average aspect ratio (length/diameter) of 2.25. Then, 0.1, 0.2, and 0.3 vol% of ZnO–EG NFs were prepared by adding the required ZnO NRs to 100 mL of EG. After that, time-lapse sedimentation observation, zeta potential (ζ), and ultraviolet-visible (UV–vis) spectroscopy was used to assess the stability of the NFs. Furthermore, the viscosity (*μ*) and density (*ρ*) of NFs were measured experimentally as a function of vol% from ambient temperature to 60 °C. Finally, the HTC of NFs was evaluated utilizing a vertical shell and tube heat transfer apparatus and a computer-based data recorder to quantify the forced convective HTC of NFs in laminar flow at Reynolds numbers (*Re*) of 400, 500, and 600. The obtained results indicate that adding only small amounts of ZnO NRs to EG can significantly increase the HTC, encouraging industrial and other heat management applications.

## 1. Introduction

Heat transfer enhancement is a significant aspect of engineering research since it is associated with heat management in industries: heat exchangers, automobiles, high-voltage transformers, refrigerators, electronics, nuclear systems, solar energy harvesters, and desalinization [1]. Water, EG, engine oil, silicone oil, gear oil, kerosene, and other conventional fluids have been used for heat transfer purposes for a long time. The poor heat transfer of conventional heat transfer fluids (e.g., the thermal conductivity of liquids lies between that of the insulators and non-metallic solids (0.1 < *k*_L_ < 10 W/mK)) has been causing problems in many sectors, including industries [2]. That heat transfer needs to be increased. Increased heat transfer can intensify some cooling processes, saving time, energy, and the amount of heat transfer equipment. Generally, adding a small number of solid particles to conventional fluids can increase heat transmission capacity remarkably [3]. However, the rapid sedimentation of bulk solids in conventional fluids is a problem in this case [4].

Nanomaterials (NMs—at least one of the dimensions is between 1 and 100 nm) are being considered to solve or mitigate the rapid sedimentation problem. NFs, the dispersion of NMs in base fluids (BFs), is one of the alternatives to conventional heat transfer fluids [5]. NFs are more stable than micron-sized materials dispersed fluids since NMs remain dispersed in a BF for a longer time [6]. Usually, metals, metal oxides, carbon nanotubes, graphene, and other NMs are used to prepare NFs to enhance the thermal conductivity by increasing the conduction and convection coefficient [7]. Especially, nano ZnO can be used to prepare NFs in conventional fluids such as EG because the thermal conductivity of ZnO (29 WmK^−1^) is more than a hundred times higher than that of ethylene glycol (0.26 WmK^−1^) [2]. Additionally, experimental results revealed that ZnO-based NFs exhibited enhanced thermal conductivity compared to other NFs [8,9,10].

BF is a key issue to consider for NFs preparation, and HCT results of NFs often depend on BFs. The literature survey provides information on using water-based [11] and oil-based NFs [12], their thermophysical properties, and their advantages and disadvantages. For instance, inside a vertical circular enclosure heated from above, the influence of an alumina-water NF on natural convection heat transfer reported that the Nusselt number of the alumina–water NF is lower than that of the BF. This suggests that, as compared to pure water, utilizing alumina–water NFs harms the HTC [13]. Therefore, choosing either an aqueous or non-aqueous BF may be worthy.

Many efforts have been made to improve the thermal conductivity of ZnO-based NFs further. Experiments were done considering the affecting factors related to the convection heat transfer of NFs. For instance, ZnO–water NFs were used in an experiment to improve convective heat transfer in a car radiator at a higher Reynolds number [14]. Heat transfer was evaluated at the laminar flow condition, and the results revealed that the heat transfer behavior of ZnO–EG-based NFs depends on the flow condition [15]. The geometric shape effect coupled with the heat exchanger of NFs in heat transfer applications was investigated and evaluated [16]. Shell and tube heat exchangers were used to experiment with the convective heat transfer enhancement of graphene NFs and the results revealed that the HTC of graphene NFs rose by 23.9% at 38 °C when the graphene concentration was increased from 0.025 to 0.1 wt% [17]. The literature review paper reveals that employing NF to improve natural convection heat transfer is still controversial, and there is an ongoing discussion about the role of NPs in heat transfer enhancement [18]. Few studies focused on the factors that affect the thermal conductivity of NFs and summarized that the dispersion stability of NFs [19], NMs’ volume%, NF preparation method, sonication time, and the morphology (size and shape) of NMs are the essential factors that affect the heat transfer properties of NFs [19,20].

The morphology, i.e., size and shape of NMs [21,22], and preparation method significantly influence the heat transfer properties of NFs. The result of a thermal conductivity dependence on ZnO-based NFs shows that the thermal conductivity increases linearly with increasing particle size [23]. An experimental investigation of the ZnO particle shape-dependent thermal conductivity of ZnO–EG NFs showed a 23% enhancement in the thermal conductivity of 0.1 mg/1 mL of ZnO NF compared to the BF (water). Again, the NF with non-spherical NPs exhibits greater thermal diffusivity than an NF with spherical NPs [24]. Moreover, instability in NFs results in a decrease in thermophysical properties as well as a reduction in heat transfer efficiency [25]. The thermal conductivity of NFs is demonstrated to be strongly influenced by the suspension stability of boehmite alumina in various shapes [26]. According to a molecular dynamic simulation, the heat transfer following the collision of cylindrical NPs with a heat source under the most favorable conditions can be substantially higher than that of spherical NPs of equal volume. As evidenced by experimental data, this can potentially result in a more significant enhancement in the thermal conductivity of an NF, including elongated particles, compared to one having spherical NPs [27]. In addition, the thermal conductivity of ZnO NFs with roughly rectangular and sphere NP morphologies was also examined experimentally at varied NP volume concentrations ranging from 0.05 to 5.0 vol% in a different study. Compared to the BF (water), the thermal conductivity of the ZnO NFs rose by 12% and 18% at 5.0 vol% for the spherical and almost rectangular forms of NPs, respectively. The shape of the particles is found to impact the increases in thermal conductivity substantially [28]. Molecular dynamics simulations were also done to investigate the effect of NPs’ aggregation morphology on the thermal conductivity of NF [29]. Moreover, the effect of the friction factor of NFs containing cylindrical nanoparticles in laminar flow was evaluated by simulation work, and the results concluded that the friction factor decreases with the increase in Reynolds numbers [30,31]. However, these papers do not include ZnO NPs. Additionally, the layering phenomenon at the liquid-solid interface in Cu- and CuO-based NFs [32], thermal conductivity based on the phonon theory of liquid [33], and clustering phenomenon in Al_2_O_3_ NFs were investigated. Results revealed that clustering reduces the thermal conductivity of NFs; however, the sedimentation phenomenon, which increases with cluster size, can cause results to alter [34]. The findings of heat transfer coefficient measurement inside Al_2_O_3_–water-filled square cuboid enclosures reveal that inclination angle is only a significant component in natural convection for enclosures with a large aspect ratio [35].

In another research, thermal conductivity measurement as a function of temperatures (10–70 °C) and ZnO NPs’ concentration (0.5–3.75 vol%) reported that, at 30 °C, a maximum thermal conductivity improvement of 40% (3.75 vol% ZnO) is found [36]. Various shapes of ZnO NMs have been used to test the thermal conductivity of ZnO–EG NFs so far. A few reports are found on the effect of NPs’ shape on the heat transfer of a shell and tube heat exchanger, and that does not include ZnO NRs [37]. According to our survey, no report is available for the study of ZnO–EG NFs with rod-shaped ZnO NPs in the laminar flow condition at low volume concentrations.

The shape of NMs has a significant impact on heat transfer performance in many applications because the specific surface area of NMs changes with the change in sizes and shapes. Moreover, NMs of the same diameter with various shapes may have different surface areas. As a result, synthesizing NMs with the desired shape is significant for improving heat transfer performance. Therefore, this study aimed to synthesize ZnO NRs and forced the convective HTC measurement of low concentration ZnO–EG NFs at laminar flow. Hence, the first task of this research is the preparation of ZnO–EG NFs.

The NF preparation procedure is significant, because it involves something beyond simply mixing the NPs with the BF. NFs can be prepared either by a two-step or a one-step method [12,38]. In the one-step method, the synthesis of NMs and the preparation of NFs are performed in a single step. In the two-step method, however, NMs are synthesized, and then the NFs are prepared by dispersing the required amount of NMs in BFs. Due to its simplicity, the two-step approach is frequently used to prepare the desired concentration. Surfactants or capping agents are often used to prepare stable NFs, and PVP surfactant has the highest degree of stability over other surfactants [39]. ZnO-based dispersion stability and thermophysical properties were assessed to evaluate the heat transfer earlier [40]. However, some limitations of water-based NFs for HTC measurement have been reported [35]. Once the NFs are prepared, stability can be assessed by observing the sedimentation and UV–vis analysis [41]. Before ZnO NRs-based NFs’ preparations and stability assessment, ZnO NRs need to be synthesized.

Therefore, synthesizing ZnO NRs by a simple and cost-effective method is essential. Physical [42], chemical [43], biological [44], and green [45] approaches are used to produce ZnO NMs. Chemical synthesis is popular because it converts a large percentage of precursors to products in a short period. Chemical precipitation, chemical vapor deposition, sol-gel, spray pyrolysis, sputtering, microwave-assisted, hydrothermal, and other synthesis methods include examples [46]. Hydrothermal synthesis has several advantages over other synthetic methods, including inexpensive water as the solvent, one-pot synthesis, low aggregation, high purity, and great control of particle size and morphology. ZnO may be synthesized at the nanoscale in a variety of shapes, including spheres [47], needles [47], flowers [48], flakes [49], tablets [50], pencils [50], and rods [51].

As the convective HTC of rod-shaped, ZnO-based NFs is not reported earlier, in this study, firstly, ZnO NRs were synthesized using a hydrothermal route for HTC measurement. Next, using ultrasonication, 0.1, 0.2, and 0.3 vol% rod-shaped ZnO–EG NFs are formulated by dispersing the required amount of ZnO NRs in EG, and their dispersion stability was assessed by time-lapse sedimentation observation, zeta potential measurement, and UV–vis analysis. Finally, employing a vertical shell and tube heat exchanger, the forced convective HTC of ZnO–EG NFs at 0.1, 0.2, and 0.3 vol% was investigated under a constant heat flux at different temperatures and laminar flow conditions of *Re* between 400 and 600.

## 2. Experimental Section

### 2.1. Materials

Zinc acetate dihydrate [Zn(CH_3_COO)_2_·2H_2_O] was supplied by Scharlau, Barcelona, Spain. Sodium hydroxide (NaOH), polyvinylpyrrolidone (PVP) as a surfactant, and ethylene glycol as a base fluid were supplied by Merck, India. All the materials were analytical grade and used without further purification. Distilled water supplied by Active Fine Chemicals. Dhaka, Bangladesh was used in all experiments.

### 2.2. Synthesis of ZnO NRs

ZnO NRs were synthesized using a slightly modified hydrothermal procedure described previously [51]. A 50 mL aqueous solution of zinc acetate dihydrate [Zn(CH_3_COO)_2_·2H_2_O] was prepared in a beaker. A 50 mL aqueous NaOH solution was prepared in another beaker by dissolving the requisite amounts of NaOH pellets into the distilled water. The 50 mL aqueous solution of NaOH was then slowly added dropwise into the precursor solution while stirring continuously. Zn(CH_3_COO)_2_·2H_2_O and NaOH had a molar ratio of 1:2. Next, 0.25 g of PVP was added to the solution mixture. After 2 h of stirring at room temperature, the mixture became milky. The solution mixtures were then transferred to a stainless-steel autoclave with a volume of 120 mL and heated in an oven at 170 °C for 6 h. The autoclave was then allowed to cool. The product was collected and rinsed three times with distilled water through centrifugation (10,100 rpm at an ambient temperature) and decantation. Finally, the product is dried in the oven at 120 °C for 3 h.

### 2.3. Characterization of ZnO NRs

An X-ray diffractometer (Phillips X’Pert PRO PW 3040, The Netherlands) using CuK radiation in a 2θ-configuration was used to analyze the crystal structure of the produced products. At a scanning rate of 0.02°/0.6 s, the scanned value of 2θ angles was between 30° and 70°. Data from the Joint Committee for Powder Diffraction Studies (JCPDS) file for ZnO were compared to the measured data (Card No. 01-070-8072). The particle size and shape of the produced ZnO NRs were examined using a transmission electron microscope (TEM, TALOS F200X, 200 KeV, The Netherlands) with an acceleration voltage of 200 kV.

### 2.4. ZnO–EG NFs Preparation

A two-step approach was used to make ZnO–EG NFs [1]. ZnO NRs were synthesized and then dispersed in EG. The loaded amount of ZnO NRs was 0.1, 0.2, and 0.3 vol% in EG. Afterward, the ZnO–EG NFs were sonicated at ambient conditions for 1 h at a frequency of 40 kHz using an ultrasonic cleaner (Model: VGT-1860QTD, China).

### 2.5. Stability Assessment of ZnO–EG NFs

The stability of the prepared 0.1, 0.2, and 0.3 vol% of ZnO–EG NFs was assessed using sedimentation observation with time interval image capturing using a digital camera, zeta potential measurement (Zeta sizer, Nano ZS, Malvern, UK), and UV–vis spectroscopy with quartz cuvettes and a path length of 10 mm using a spectrophotometer (UV-1800, Shimadzu, Japan). The measurement was performed in the wavelength range from 200 to 800 nm. Absorbance was taken at 1, 2, 3, and 4 h for 0.1, 0.2, and 0.3 vol% ZnO–EG NFs and plotted as a function of wavelength. UV–vis spectrums were compared with the stability change of the NFs with time.

### 2.6. Viscosity Measurements of ZnO–EG NFs

Viscosity (*μ*) of the BF, i.e., 0% and 0.1, 0.2, and 0.3 vol% ZnO–EG NFs, was measured using a DV-II + Pro EXTRA, BROOKFIELD VISCOMETER, the USA, at ambient temperature and 40, 50, and 60 °C, and 100 rpm following the user’s manual. A total of 250 mL NFs was needed for viscosity measurement using a spindle. The uncertainty of the temperature control is ±5 °C.

### 2.7. Density Measurements of ZnO–EG NFs

The density (*ρ*) of the BF and 0.1, 0.2, and 0.3 vol% ZnO–EG NFs were measured using a Pycnometer. The 50 mL samples were used for measurements. The weight of the blank Pycnometer was taken at ambient temperature, 40, 50, and 60 °C. Similarly, the weights of the pycnometer with BF, 0.1, 0.2, and 0.3 vol% ZnO–EG NFs were taken at ambient temperature, 40, 50 and 60 °C. Next, the density was calculated by subtracting the weight and dividing it by mass. The uncertainty of the balancing is 0.001 g.

### 2.8. HTC Measurement of ZnO–EG NFs

An experimental setup was established to determine the convective HTC of the NFs. Figure 1 schematically depicts the experimental setup. Previously, this kind of setup was used for HTC measurements using CuO−PVA NFs [52]. The experimental setup consisted of a 350 mL reservoir shell in which hot water was kept at a constant temperature of 100 °C. Moreover, hot water was kept at a constant temperature of 40, 50, and 60 °C for some measurements. The water is heated by 2 kW immiscible heaters (Watlow ref. L14JX8B) installed in the shell. The heat flux was maintained at a constant rate. Incorporated inside the shell was a spiral tube constructed of smooth copper tubing with a 0.44 × 10^−3^ m outer diameter, 0.15 × 10^−3^ m thickness, 0.29 × 10^−3^ m inner diameter, and a heat exchange length of 0.6858 m. The NFs and BF were picked up into a reservoir using a centrifugal pump (Espa Ref. XVM8 03F15T). The Reynolds number of the flow was between 400 and 600. Then, the NFs and BFs were passed through the test area. Laminar flow was confirmed by trialing the flow of ZnO–EG NFs. Some pigments were added in the ZnO–EG, and the flow was observed. No vertical flow of pigments was observed or no layer change of pigments was observed. In this way, it was characterized that the flow was laminar. The fluids absorbed the heat from the hot water and then exited the tube. Hot fluids were collected in a beaker as they traveled through the tube. The test area was appropriately insulated with glass wool to reduce heat loss from the system. The temperature of the shell and outflow side tube is measured using K-type thermocouples.

The heat flux determines the HTC via a single wall surface and heat conductivity across the boundary layer over the wall surface is required to complete heat transfer. With a change in wall temperature, the HTC is proportionality constant between the heat supplied and the thermodynamic driving force of heat flow through the per unit area. The spatial molecular diffusion of heat throughout the fluid is related to the thermal conductivity of the fluid [52]. The NFs were used within the stable period of 2 h from their preparation. The experiments were repeated at least three times and the average value was taken as the HTC coefficient. Standard deviation was included by adding an error bar in the column graph.

## 3. Results and Discussion

### 3.1. Characterization of Synthesized ZnO NRs

Analysis of the XRD peak pattern of the product can reveal the crystalline nature and phase structure of the product. Figure 2 depicts the XRD pattern of powder-type ZnO. The corresponding diffractogram in the 2-theta range of 30–70° indicated ZnO’s hexagonal wurtzite phase (b), corresponding to the standard card [Card No. 01-070-8072] (a). No other peaks due to contamination were observed. Thus, the single-phase structure of ZnO has successfully synthesized with a {100} plane and the results were similar to other previous reports [15,16].

The morphology of ZnO was examined using TEM analysis. Figure 3 shows the TEM image of the synthesized ZnO, which shows the rod-shaped particles. As shown in Figure 3a, different sizes of ZnO rods were created with different diameters and lengths. The smallest diameter of a single ZnO NR is estimated at 50 nm, indicating the production of nanoscale ZnO NRs. Figure 3b shows the absence of rod branching, indicating that the ZnO NRs were developed from spontaneous nucleation with excellent crystal perfection [53]. Of course, their wide size distribution is seen in the images. The ZnO NR’s aspect ratio was calculated by dividing the length by the diameter of the ZnO NR. In many applications, the aspect ratio of rod-shaped particles is significant, as the NRs with a higher aspect ratio provide larger specific surface areas than the NRs with a smaller aspect ratio. The NRs’ average length and diameter are 270 and 120 nm, respectively, and the mean aspect ratio of ZnO NRs was found to be 2.25.

### 3.2. Stability Assessment of ZnO–EG NFs

First, the sedimentation observation method was performed for the stability test of NFs [54]. All the prepared NFs appeared to be white or whitish initially, and a slight color change occurred as sedimentation increased with increasing time intervals. Figure 4 shows the time-lapse observation during the sedimentation procedure of NFs. As seen in Figure 4a,b, no sedimentation is observed for all three samples. As the time increased, however, ZnO NRs tended to settle down to the bottom of the container due to sedimentation. Figure 4c,d shows that some sedimentation was deposited at the bottom of the bottles at 3 and 4 h, especially for the 0.2 vol% and 0.3 vol% NF samples. In the case of 4 h (Figure 4d), the upper dispersion phase becomes slightly transparent, showing more sedimentation at the bottom of the bottles. Therefore, it is considered that the prepared NFs are stable for at least 2 h. Considering the instability and sedimentation, HTC experiments were conducted within the stability period of NFs.

Agglomeration and settlement of NPs in BFs greatly contribute to channel clogging and affect thermal conductivity. Thus, stability is one of the most important factors affecting NF quality. Zeta potential is an important tool to assess the stability of NFs. In NF stability, the zeta potential of (+60 mV) to (−60 mV) is generally used as a border value. NFs having zeta potential values of more than +60 mV or less than −60 mV are often stable [55]. However, zeta potentials of between +10 mV and −10 mV or close to zero indicate the instability of NFs [56]. Table 1 shows the zeta potential of prepared NFs. The zeta potentials of 0.1, 0.2, and 0.3 vol% ZnO–EG NFs at 2 h of their preparation were −26.3 mV, 13.5 mV, and 4.1 mV, respectively. According to zeta potential measurements, the stability of the prepared NFs decreases with the increase in concentration. Moreover, 0.1 and 0.2 vol% ZnO–EG NFs are within the stability range, indicating a homogenous dispersion and good stability. However, 0.3 vol% ZnO–EG NF is out of the stability range.

Next, UV–vis analysis was performed on the NFs samples to assess the stability further. This is because the state and stability of the NFs can be evaluated by the difference in UV–vis absorption wavelength [57]. Here, the UV–vis spectra of ZnO–EG NFs were measured at 1, 2, 3, and 4 h of their preparation. Figure 5 shows the UV–vis spectrum of ZnO–EG NFs at 0.1 vol% (a), 0.2 vol% (b), and 0.3 vol% (c) at different times between 1 and 4 h. Figure 5a shows that the absorbance of NFs at a wavelength of 800 nm decreases harmonically with the increase in time. The spectrum pattern for 1, 2, and 3 h is similar, except for that of 4 h. The absorbance at 4 h significantly decreased from the initial state (1 h), which indicates that the 0.1 vol% ZnO–EG NFs is stable up to 3 h after their preparation. The drastic change in the absorbance pattern indicates that agglomeration and sedimentation occurred rapidly after 3 h of its preparation.

Figure 5b shows the UV–vis spectra of 0.2 vol% ZnO–EG NFs. The absorbance of 0.2 vol% ZnO–EG NFs at the wavelength of 800 nm decreases with the increase in time. Generally, absorbance values of 0.2 vol% ZnO–EG NFs at 800 nm are lower than that of 0.1 vol% ZnO–EG NFs, indicating a relatively low concentration of the dispersed phase. Although this phenomenon could not be accurately confirmed with a photograph, it can be considered reasonable because it coincides with the change in the zeta-potential value. It can be seen that absorbance after 2 h becomes lower fast; this is probably because unstable particles were removed by precipitation due to the time-lapse of 0.2% ZnO–EG NFs. Thus, Figure 5b indicates that the 0.2% ZnO–EG NFs are stable until 2 h of its preparation.

Figure 5c shows the UV–vis spectrum of 0.3 vol% ZnO–EG NFs. Similar to Figure 5b, the initial absorbance value decreased with an increase in time. Here, the absorbance at 800 nm decreased quickly when the 2 h time had passed. Finally, it can be said that 0.1 vol% ZnO–EG NF is stable for up to 3 h, and 0.2 and 0.3 vol% ZnO–EG NFs are stable for at least 2 h.

### 3.3. Viscosity of ZnO–EG NFs 

Average viscosity of BF (EG) and 0.1, 0.2, and 0.3 vol% ZnO–EG NFs at ambient temperature and 40, 50, and 60 °C at 100 rpm were found to be 7.40, 7.60, 7.85, and 8.46 cP, respectively, as shown in Table 2. Relative viscosity was also calculated, which indicates no significant differences at different concentrations of NFs. It was found that adding NPs to NFs increased their viscosity by a tiny amount, making them suitable for heat transfer applications with minimal pressure drop in the flow channels. Viscosity values are important to understand the stability of NFs. Moreover, viscosity data are needed to calculate the Reynolds number of the flow of NFs.

### 3.4. Density of ZnO–EG NFs

The density of ZnO–EG NFs was calculated using the procedure described in the experimental section. Density is another significant parameter of NFs that is needed to know the Reynolds number of flowing NFs. Table 3 shows the calculation of the density of NFs. Data shows that the minimal density occurs due to adding a small amount of ZnO NRs to EG.

### 3.5. HTC of ZnO–EG NFs

Convective heat transfer between a moving fluid and a solid surface can be defined by the following relationship [58]:(1)$$Q=hA\;\left(T_{s}-T_{f}\right)=hA\Delta T$$
where *Q* is the rate of forced convection heat transfer (W), *T*_s_ is the solid surface temperature (K), *T*_f_ is the fluid temperature (K), *A* is the area of the surface that is in contact with the fluid (m^2^), *h* is the convective HTC (W/m^2^·K).
*Q* = *mS*Δ*θ*(2)
where *m* is the mass of hot water, kg, *S* is the specific heat of water, J/kg·K, Δ*θ* is the difference in water temperature before and after releasing temperature, K.

These relationships can be summarized as follows:(3)h=(Q/A)/(ΔT)=mSΔT/(A·ΔT)

Let the HTC of the 0 (BF), 0.1, 0.2, 0.3 vol% ZnO–EG NFs be *h*_0_, *h*_1_, *h*_2_, *h*_3_, respectively. The temperature difference before and after release temperature was set as Δ*θ*_0_, Δ*θ*_1_, Δ*θ*_2_, and Δ*θ*_3_ for the 0 (BF), 0.1, 0.2, and 0.3 vol% ZnO–EG NFs, respectively. After absorbing heat, the temperature difference between H_2_O and each NFs is Δ*T*_0_, Δ*T*_1_, Δ*T*_2_, and Δ*T*_3_, corresponding to 0 (BF), 0.1, 0.2, and 0.3 vol% ZnO–EG NFs, respectively. All the parameters for the calculation are summarized in Table 4.

Forced convective HTC of BF and ZnO–EG NFs at 0 (BF), 0.1, 0.2, and 0.3 vol% were measured three times under laminar flow conditions. Obtained data is summarized and presented in Table 5.

Average values of forced convective HTC of 0 (BF), 0.1, 0.2, and 0.3 vol% of ZnO–EG NFs were calculated as 219, 1284, 2156, and 2536 Wm^2^/K, respectively. Results were plotted in Figure 6 with a standard error bar from standard deviations. As seen from the column graph, the HCT values were increased as the concentration of NRs was increased. The HTC of 0.1, 0.2, and 0.3 vol% NFs were 6, 10, and 12 times higher than that of BF, respectively, indicating the non-linearity of HTC increment. The results indicated that small amounts of ZnO NRs could greatly enhance their HTC values. Notably, 0.1 vol% showed that the best HTC and 0.3 vol% have the lowest HTC compared to BF.

The sharp increase in HTC due to the input of the nanorods is apparent, and it can be seen that it is related to the concentration of the NRs in BFs. However, HTC did not increase linearly to the increments of the NRs’ concentration. This phenomenon strongly depends on the stability of NFs. The previous section shows that relatively high concentrated NFs (e.g., 0.3%) showed unstable conditions. Thus, local aggregation or agglomeration of ZnO NRs may occur during the HTC measurements. The tendency of the HTC results is consistent with the results of UV–vis shown in Figure 5.

Again, forced convective HTC was measured by varying the temperature of hot water. Figure 7 shows the HTC of 0.1, 0.2, and 0.3 vol% ZnO−EG NFs at different temperatures with standard deviations. If the source of heat (hot water) is kept at higher temperatures, the heat transfer is higher as seen in Figure 7. Average HTC of 2563, 3390, and 3686 were calculated for the hot water temperature of 40 °C, 50 °C, and 60 °C. Therefore, it seems significant to maintain the temperature of heat source fluid at an optimum higher degree.

Another important factor for heat transfer fluid flow is Reynolds numbers (*Re*), which need to be considered during NF flow. Figure 8 presents the forced convective HTCs of 0.1, 0.2, and 0.3 vol% ZnO−EG NFs at different Re. Experiments were performed Re at 400, 500, and 600. Data indicate that the HTC of the same concentration of NFs increases with the increase in Reynolds numbers.

## 4. Conclusions

The forced convective HTCs of BF (EG) and 0.1, 0.2, and 0.3 vol% ZnO–EG NFs were investigated in this study. For this purpose, ZnO NRS was successfully synthesized by the hydrothermal method. The prepared ZnO was hexagonal wurtzite and had a nano-sized rod shape with an average aspect ratio (length/diameter) of 2.25. ZnO–EG NFs at 0.1, 0.2, and 0.3 vol% were prepared by dispersing ZnO NRs to EG with the aid of 1 h ultrasonication. Time-lapse sedimentation observation, zeta potential measurement, and UV–vis analysis indicated that all the prepared NFs were stable at least for 2 h. Among the NFs, 0.1 vol% ZnO–EG NFs showed the best stability within 3 h. The HTC of ZnO–EG NFs was significantly increased with low loading concentration of ZnO NRs, and those HTCs of 0.1, 0.2, and 0.3 vol% NFs increased for 6, 10, and 12 times compared to BF (EG). At a high ZnO concentration, the HTC was not increased linearly with the increments of NRs’ concentration, probably due to the local agglomeration during the measurements. Again, a higher Reynolds number leads to the higher HTC of an NF compared to that of a lower Reynolds number. Therefore, we expect that the well-dispersed ZnO concentration in NFs will significantly increase the HTC in NFs; this will facilitate ZnO–EG NFs in many industrial and other thermal management applications. By selecting a stabilizer under optimal ultrasonication time for enhanced stability, the current study scheme can be explored further to manufacture similar types of aqueous/non-aqueous metal oxide NFs. Moreover, at higher Reynolds, the HTC of current NF systems may need to be investigated further.

## Figures and Tables

**Figure 1 nanomaterials-12-01568-f001:**
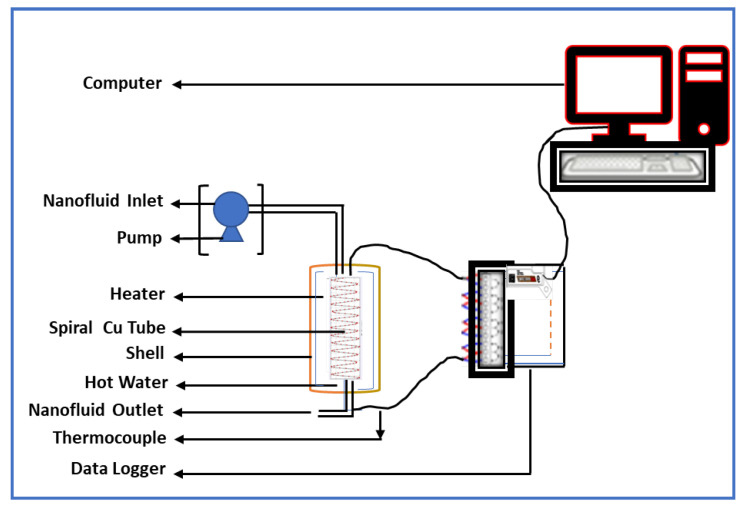
Experimental setup for the measurement of forced convective HTC of ZnO–EG NFs.

**Figure 2 nanomaterials-12-01568-f002:**
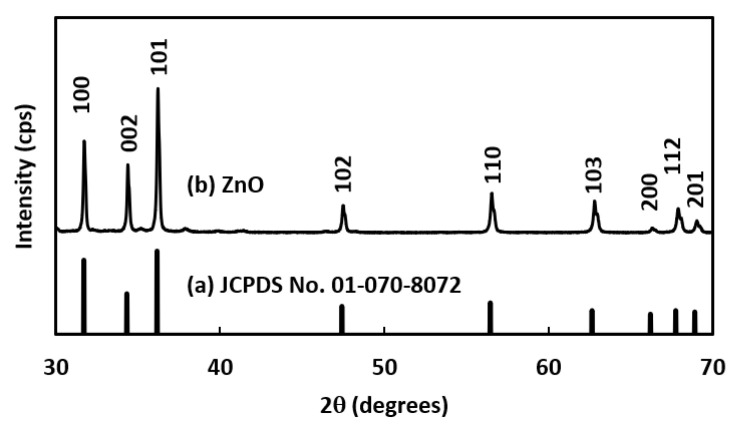
XRD peak pattern of the products: (**a**) JCPDS and (**b**) Synthesized ZnO NRs.

**Figure 3 nanomaterials-12-01568-f003:**
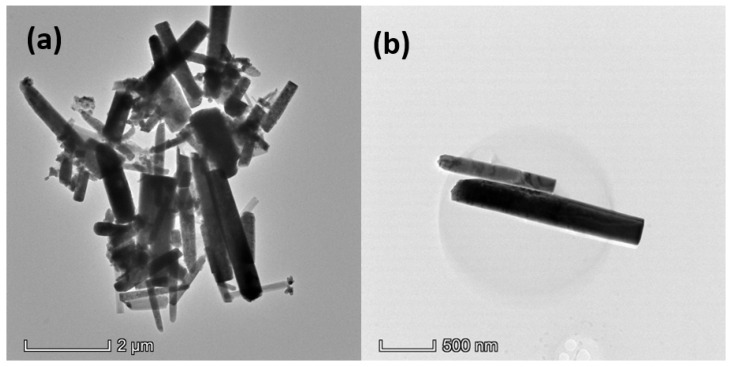
TEM images of the synthesized ZnO NRs using the hydrothermal method: (**a**) images at 2 µm scale with the magnification of 7000 times and (**b**) images at 500 nm scale with the magnification of 17,500 times.

**Figure 4 nanomaterials-12-01568-f004:**
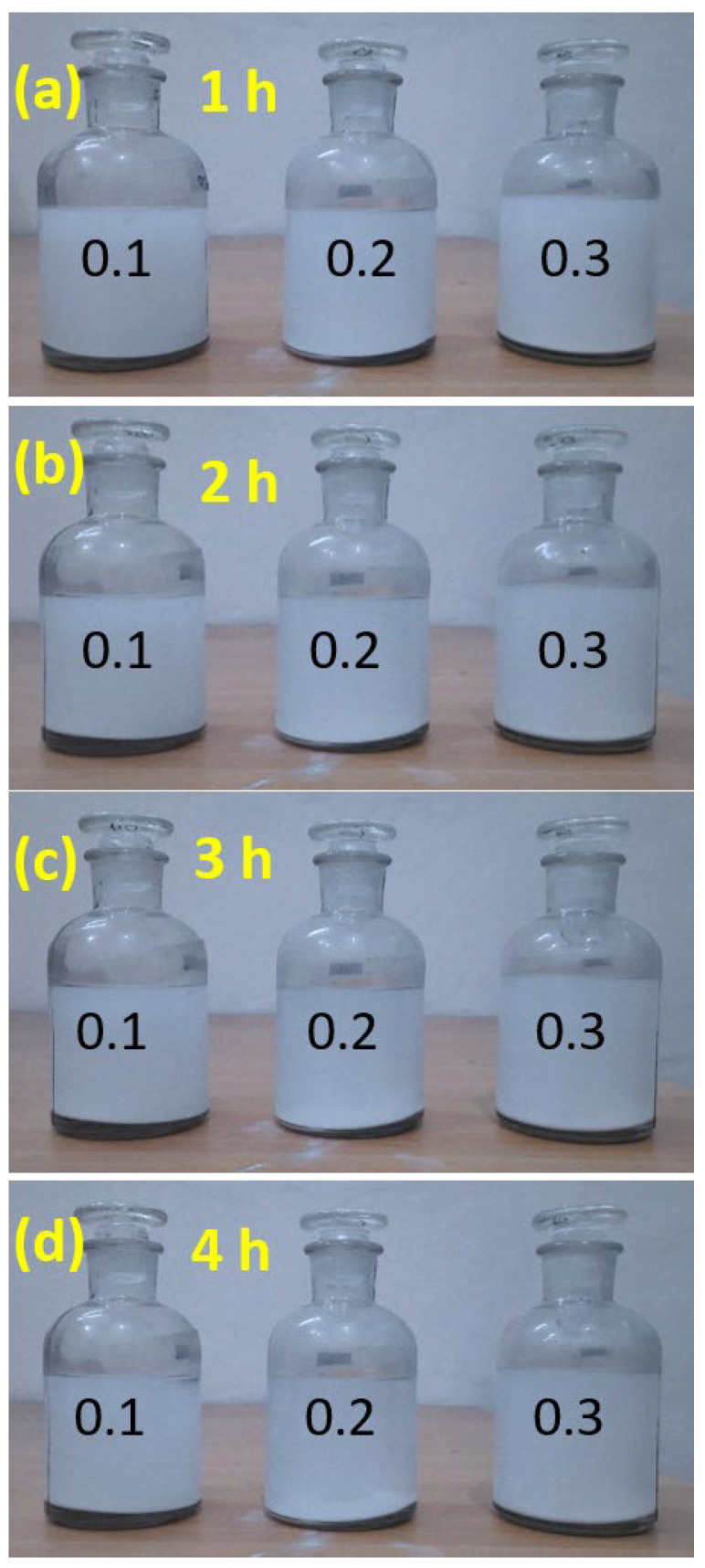
Time-dependence sedimentation photo images of ZnO–EG NFs (0.1, 0.2, 0.3 vol%) after the NFs preparation: (**a**) 1 h; (**b**) 2 h; (**c**) 3 h; (**d**) 4 h.

**Figure 5 nanomaterials-12-01568-f005:**
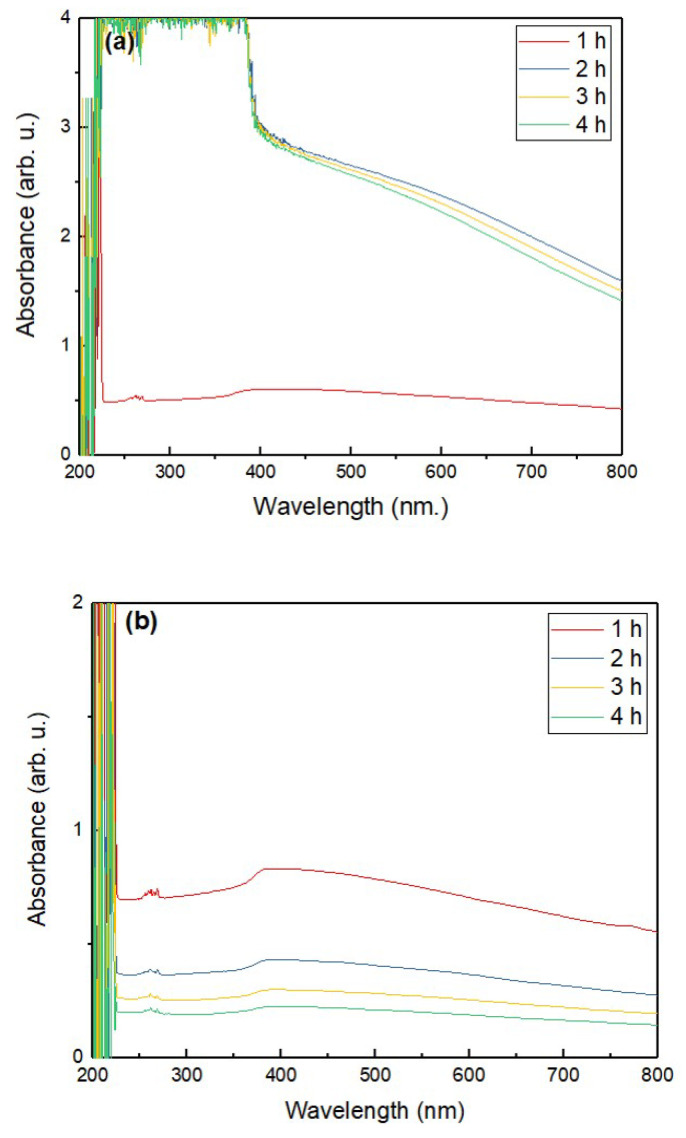
UV–vis spectra of ZnO–EG NFs according to the different time span between 1 and 4 h at different volume percent: (**a**) 0.1 vol%, (**b**) 0.2 vol%, and (**c**) 0.3 vol% of ZnO.

**Figure 6 nanomaterials-12-01568-f006:**
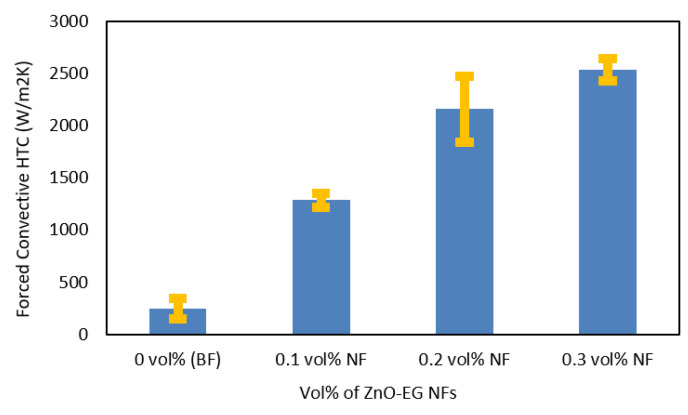
Forced convective HTC of BF and ZnO–EG NFs at different vol% of NRs at room temperature.

**Figure 7 nanomaterials-12-01568-f007:**
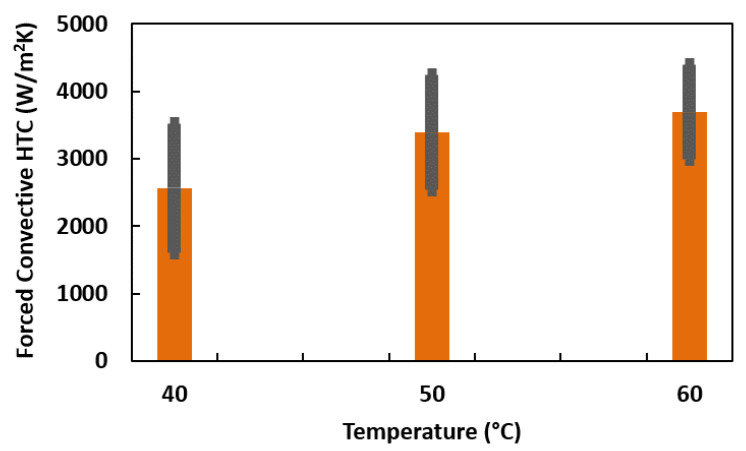
Forced convective HTC of 0.1, 0.2, and 0.3 vol% ZnO–EG NFs at different temperatures.

**Figure 8 nanomaterials-12-01568-f008:**
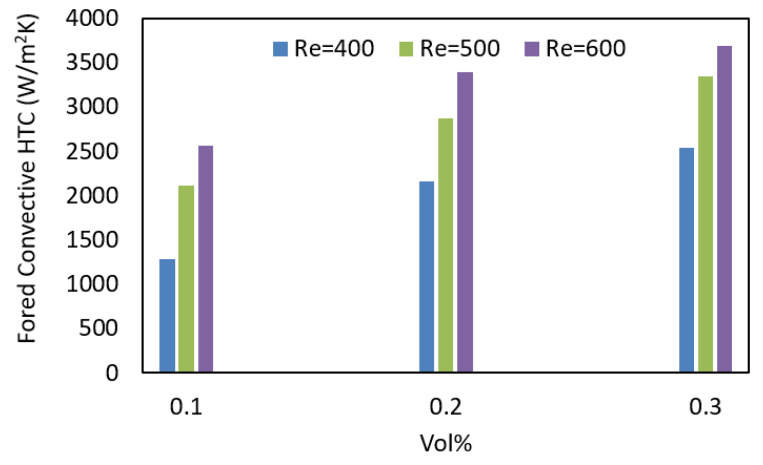
Forced convective HTCs of 0.1, 0.2, and 0.3 vol% ZnO–EG NFs at different Reynolds numbers.

**Table 1 nanomaterials-12-01568-t001:** Zeta potential of 0.1, 0.2, and 0.3 vol% ZnO–EG NFs at 2 h of their preparation.

Concentration (vol%)	Zeta Potential, ζ (mV)
0.1	−26.3
0.2	13.5
0.3	4.1

**Table 2 nanomaterials-12-01568-t002:** The viscosity of BF (EG) and 0.1, 0.2, and 0.3 vol% ZnO–EG NFs at ambient temperature and 100 rpm.

Concentration (vol%)	Viscosity, *μ* (cP)	Relative Viscosity, *μ_r_*
0 (EG)	7.40	-
0.1	7.60	1.02
0.2	7.85	1.06
0.3	8.46	1.14

**Table 3 nanomaterials-12-01568-t003:** Density of BF (EG) and 0.1, 0.2, and 0.3 vol% ZnO–EG NFs at ambient temperature, 40, 50, 60 °C.

Sample Conc. (vol%)	Ave. Weight of BF and NFs at Different Temperatures, *W*_2_				Ave. wt. of Blank Pycnometer (50 mL), *W*_1_	Density, *ρ* (g/m^3^) = (*W*_2_ − *W*_1_)/50
	Ambient Temperature	40 °C	50 °C	60 °C		
0 (EG)	73.30	72.99	72.77	72.76	26.67	0.932
0.1	73.57	73.31	73.03	72.51	26.67	0.933
0.2	73.25	73.24	73.25	73.28	26.67	0.938
0.3	73.85	73.52	73.27	72.83	26.67	0.944

**Table 4 nanomaterials-12-01568-t004:** Summary of the parameter for HTC of ZnO–EG NFs.

Parameter	Value
Mass of water	0.35 kg
Specific heat of water, *S*	4178 J/kg·K
The outer diameter of the Cu-tube, *d*	4.4 × 10^−4^ m
Heat exchange length of Cu-tube, *l*	68.58 × 10^−2^ m
Δ*θ*_0_	0.01 K
Δ*θ*_1_	1.3 K
Δ*θ*_2_	1.3 K
Δ*θ*_3_	1.2 K
Δ*T*_0_	13.49 K
Δ*T*_1_	4.5 K
Δ*T*_2_	5.9 K
Δ*T*_3_	4.1 K

**Table 5 nanomaterials-12-01568-t005:** Forced convective HTC of BF and ZnO–EG NFs at 0.1, 0.2, and 0.3 vol% at *Re* = 400.

Concentration (vol%)	Forced Convective HTC (Wm^2^/K)		
	1st Measurement	2nd Measurement	3rd Measurement
0 (BF)	144	292	220
0.1	1341	1212	1299
0.2	2195	2449	1823
0.3	2656	2452	2500

## Data Availability

Not applicable.

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
