# Peer review of "Forced Convective Heat Transfer Coefficient Measurement of Low Concentration Nanorods ZnO–Ethylene Glycol Nanofluids in Laminar Flow"

_nanomaterials, 2022, doi:10.3390/nano12091568_

Round 1
Reviewer 1 Report
The paper deals with experiments on the behavior of Nanorods to increase the heat transfer coefficient. After an introduction to the problem, the paper describes how the nanorods are synthesized and the subsequent experiments.
The paper has aspects of interest but some obscure points are present that require clarification.
First of all the number of experiments for each of the concentrations seems limited and therefore the estimators are too random. Moreover, the Reynolds number 1800 is typical of a transition regime from laminar to turbulent and therefore the results must be interpreted in the light of this indication.
It is also necessary to analyze the case where the regime is turbulent, since significant differences may arise in efficiency.
A second critical point concerns the rheological behavior of the suspension. It is well known that homogeneous two-phase fluid systems flow with segregation: this is the case, for example, of blood, where erythrocytes are not uniformly distributed in the cross section of vessels, and this enhances the non-Newtonian behavior of the fluid. This part should be implemented.
In the description, see l.331 and 332, reference is made to four different values: it must be explicitly stated that also the 0 concentration is considered.
Minor comments:
units must be SI, inches is not admitted.
Author Response
We gratefully acknowledge the constructive comments and valuable suggestions of the reviewer and the editor. We have read the reviewer’s comments carefully and tried our best to revise the manuscript accordingly. The responses to the reviewer’s comments are listed below. In the revised manuscript, the revised/corrected parts were highlighted using the “Track Changes” function.
Point 1: The paper deals with experiments on the behavior of nanorods to increase the heat transfer coefficient. After an introduction to the problem, the paper describes how the nanorods are synthesized and the subsequent experiments.
The paper has aspects of interest, but some obscure points are present that require clarification.
First of all, the number of experiments for each of the concentrations seems limited, and therefore the estimators are too random. Moreover, the Reynolds number, 1800, is typical of a transition regime from laminar to turbulent, and therefore the results must be interpreted in the light of this indication.
Response 1: We would like to thank the reviewer for reviewing our paper and commenting on the aspect of interest in the present research which is inspiring to us. Required clarification for some points of the comments is given below:
We have increased the number of experiments. We have designed and performed a series of experiments related to HTC. For instance, the HTC was measured at different Reynolds numbers of 400, 500, and 600. For each concentration (0.1, 0.2, 0.3 vol%), three measurements were made and the average values were taken for the calculation. Also, we have measured the HTC at different temperatures of 40 °C, 50 °C, and 60 °C other than ambient temperature. As the number of experiments for each of the concentrations increased earlier, therefore, the estimators are not random now.
Point 2: Moreover, the Reynolds number, 1800, is typical of a transition regime from laminar to turbulent, and therefore the results must be interpreted in the light of this indication.
Response 2: The Reynolds number for measuring the forced convective HTC falls between 400 and 600 (not 1800, and we checked it and corrected it in respective palaces in the manuscript). Reynolds numbers 400-600 do not fall in the turbulent flow. The flow is laminar.
Point 3: It is also necessary to analyze the case where the regime is turbulent, since significant differences may arise in inefficiency.
Response 3: As we explained in the previous point, the Reynolds number ranges between 400 and 600, and that is laminar flow, not turbulent flow. These Reynolds numbers are quite low and do not fall during the regime of turbulence. Flows are laminar only.
Point 4: A second critical point concerns the rheological behavior of the suspension. It is well known that homogeneous two-phase fluid systems flow with segregation: this is the case, for example, of blood, where erythrocytes are not uniformly distributed in the cross-section of vessels, and this enhances the non-Newtonian behavior of the fluid. This last part should be implemented.
Response 4: Although interesting, the rheological behavior of suspension is out of the scope of the present study. We hope to implement the rheological behavior of such NFs in our future projects. However, with the limited scope, we measured the viscosity of NFs and discussed it in the manuscript. Viscosity is needed to calculate the Reynolds number of the flow of NFs. Table 2 represents the viscosity of NFs.
Table 2. The viscosity of BF (EG), and 0.1, 0.2, and 0.3 vol% ZnO–EG NFs at ambient temperature and 100 rpm.
|
Concentration (vol%) |
Viscosity, μ (cP) |
Relative Viscosity, μr |
|
0 (EG) |
7.40 |
- |
|
0.1 |
7.60 |
1.02 |
|
0.2 |
7.85 |
1.06 |
|
0.3 |
8.46 |
1.14 |
Also, we believe that segregation does not occur in NFs. Sedimentation occurs for a long time since the NRs are not surface capped. As a consequence, agglomeration occurs and particles are sedimented.
Point 5: In the description, see l.331 and 332, reference is made to four different values: it must be explicitly stated that the 0 concentration is also considered.
Response 5: In description, section 1, 331 and 332, it is mentioned that 0 concentration (i.e. only base fluid) was considered with 0.1, 0.2, and 0.3 vol% of ZnO NFs. It was added in Figure 6 and its explanation in the text is as follows:
Average values of forced convective HTC of 0 (BF), 0.1, 0.2, and 0.3 vol% of ZnO–EG NFs were calculated as 219, 1284, 2156, and 2536 Wm2/K, respectively. Results were plotted in Figure 6 with a standard error bar from standard deviations. As seen in the column graph, the HCT values increased as the concentration of NRs increased. The HTC of 0.1, 0.2, and 0.3 vol% NFs was 6, 10, and 12 times higher than that of BF, respectively. The results indicated that small amounts of ZnO NRs could greatly enhance their HTC values. More elaborately, it can be said that 0.1 vol% has the highest HTC of 6 times than EG whereas that of 0.2 vol% is 10 times and 0.3 vol% is 12 times; indicating non-linearity of HTC increment. Notably, 0.1 vol% showed the best HTC and 0.3 vol% have the lowest HTC compared to BF.
Figure 6. Forced convective HTC of BF and ZnO–EG NFs at different vol% of NRs at room temperature.
Point 5: Minor comments: units must be SI, and inches are not admitted.
Response 5: SI units were used throughout the revised manuscript.

Reviewer 2 Report
The paper "Forced Convective Heat Transfer Coefficient Measurement of Low Concentration Nanorods ZnO–Ethylene Glycol NFs in Laminar Flow" deals with an interesting subject, but it needs some revisions:
1) The authors should carefully revise the manuscript for the language.
2) A nomenclature should be added in order to list all the symbols and the corresponding units of measure.
3) A deeper bibliographic analysis is required in order to give a complete understanding of the state of the art. It is suggested to add these papers (and more):
An investigation of layering phenomenon at the liquid-solid interface in Cu and CuO based nanofluids, (2016) International Journal of Heat and Mass Transfer, Volume 103, pp. 564-571;
An explanation of the Al2O3 nanofluid thermal conductivity based on the phonon theory of liquid, (2016) Energy, 116, pp. 786-794
Thermal conductivity, viscosity and stability of Al2O3-diathermic oil nanofluids for solar energy systems, Energy, Volume 95, 2016, Pages 124-136, DOI: 10.1016/j.energy.2015.11.032
A critical analysis of clustering phenomenon in Al2O3 nanofluids, 2019, Journal of Thermal Analysis and Calorimetry 135(1), pp. 371-377, DOI: 10.1007/s10973-018-7099-9
4) More details should be given on how the thermophysical properties of the nanofluids are accounted for (thermal conductivity etc.).
5) More comments explaining the physical effects which determined the obtained results are required. Only descriptions of the plots are presented, with limited physical explanation.
6) Hence, I believe that the authors have to redraft the manuscript with added discussions providing significant physical insights.
Author Response
We gratefully acknowledge the constructive comments and valuable suggestions of the reviewer. We have read the reviewer’s comments carefully and tried our best to revise the manuscript accordingly. The responses to the reviewer’s comments are listed below. In the revised manuscript, the revised/corrected parts were highlighted using the “Track Changes” function.
According to reviewer comments, the paper "Forced Convective Heat Transfer Coefficient Measurement of Low Concentration Nanorods ZnO–Ethylene Glycol NFs in Laminar Flow" deals with an interesting subject, but it needs some revisions:
The revisions are given below:
Point 1: The authors should carefully revise the manuscript for the language.
Response 1: Our sincere thanks to the reviewer for the comments to improve the language of the manuscript. In the revised manuscript, we have checked the English grammar and language to improve it further to our best.
Point 2: A nomenclature should be added to list all the symbols and the corresponding units of measure.
Response 2: A nomenclature was added in the revised version of the manuscript after the conclusion section that lists all the abbreviations, symbols, and the corresponding units of measure.
Point 3: A deeper bibliographic analysis is required to give a complete understanding of the state of the art. It is suggested to add these papers (and more):
An investigation of layering phenomenon at the liquid-solid interface in Cu and CuO based nanofluids, (2016) International Journal of Heat and Mass Transfer, Volume 103, pp. 564-571;
An explanation of the Al2O3 nanofluid thermal conductivity based on the phonon theory of liquid, (2016) Energy, 116, pp. 786-794
Thermal conductivity, viscosity and stability of Al2O3-diathermic oil nanofluids for solar energy systems, Energy, Volume 95, 2016, Pages 124-136, DOI: 10.1016/j.energy.2015.11.032
A critical analysis of clustering phenomenon in Al2O3 nanofluids, 2019, Journal of Thermal Analysis and Calorimetry 135(1), pp. 371-377, DOI: 10.1007/s10973-018-7099-9
Response 3: We have gone through the deeper bibliographic analysis and added 8 new recent and relevant references (Ref. No. 11, 12, 13, 32, 33, 34, 35, and 40), including the ones mentioned above. We have also rearranged the numbering of references accordingly.
Point 4: More details should be given on how the thermophysical properties of the nanofluids are accounted for (thermal conductivity etc.).
Response 4: We measured the heat transfer coefficient of ZnO−EG NFs at different concentrations. Detailed explanations are added in the experimental section.
Point 5: More comments explaining the physical effects which determined the obtained results are required. Only descriptions of the plots are presented, with a limited physical explanation.
Response 5: A new explanation and description are added in the revised manuscript.
.
Point 6: Hence, I believe that the authors have to redraft the manuscript with added discussions providing significant physical insights.
Response 6: We have redrafted the manuscript by adding a series of new experiments. Every measurement was performed at least 3 times, and the average value was taken for calculations. Figures (Figure 7 and Figure 8), tables (Table 2 & Table 3) were added with their explanations. We have checked the English language and grammar to our best. A list of tables was also added after the conclusion section. Only SI units were used in the new version of the manuscript draft. Recent and relevant new citations (a total of 8) were added based on the literature review. We hope the drafted manuscript is now is an improved version.

Reviewer 3 Report
Please see attached file.

Author Response
We gratefully acknowledge the constructive comments and valuable suggestions of the reviewer. We have read the reviewer’s comments carefully and tried our best to revise the manuscript accordingly. The responses to the reviewer’s comments are listed below. In the revised manuscript, the revised/corrected part was highlighted using the “Track Changes” function.
Point 1: This paper does not have enough scientific data about forced convection heat transfer as the title said, therefore it cannot be accepted. Major work must be done as in the
following recommendation below:
This article presents an experimental investigation of Forced Convective Heat Transfer Coefficient Measurement of Low Concentration Nanorods ZnO–Ethylene Glycol Nanofluids in Laminar Flow. While the article focused more on the characterization and stability of nanofluids, the results related to convection heat transfer are very limited, as only figure 6 is included in this article. The introduction part and some experiments should be added, as well as the calculations of uncertainties should also be added with HTC of ZnO–EG NFs results in part. The following comments must be addressed to improve the article:
The literature review presented ZnO nanfluid or ZnO as nanoparticles and the article explained their importance, thermal conductivity, and applications, but the significant contribution of this article is the application in heat transfer, the article does not include enough references in this section. It is possible to bring up some papers that talk about water as a base fluid, and some papers that contain Ethylene Glycol as base fluid, and talk about their results in heat transfer, then present the gap at the end of the literature.
Response 1: We would like to thank again the reviewer sincerely for the scholastic comments.
First of all, we have done a series of new experiments considering the Reynolds number, temperatures, viscosity, density, etc. Every measurement was done at least 3 times and the average values were taken for the calculation. Those scientific data were added in the revised version of the manuscript. For instance, viscosity (Table 2) and density (Table 3) of NFs, HTC dependence of temperature by measuring at 40, 50, and 60 °C (Figure 7), and Reynolds number (400, 500, and 600) were added (Figure 8). These were added to the manuscript as follows:
3.3. Viscosity of ZnO–EG NFs
Average viscosity of BF (EG), and 0.1, 0.2, and 0.3 vol% ZnO–EG NFs at ambient temperature, at 40, 50, and 60 °C, and 100 rpm were found to be 7.40, 7.60, 7.85 and 8.46 cP respectively as shown in Table 2. Relative viscosity was also calculated which indicates no big differences at different concentrations of NFs. It was found that adding NPs to NFs increased their viscosity by a tiny amount, making them suitable for heat transfer applications with minimal pressure drop in the flow channels. Viscosity values are important to get an idea about the stability of NFs. Also, viscosity data is needed to calculate the Reynolds number of the flow of NFs.
Table 2. The viscosity of BF (EG), and 0.1, 0.2, and 0.3 vol% ZnO–EG NFs at ambient temperature and 100 rpm.
|
Concentration (vol%) |
Viscosity, μ (cP) |
Relative Viscosity, μr |
|
0 (EG) |
7.40 |
- |
|
0.1 |
7.60 |
1.02 |
|
0.2 |
7.85 |
1.06 |
|
0.3 |
8.46 |
1.14 |
3.4 Density of ZnO–EG NFs
The density of ZnO–EG NFs was calculated using the procedure described in the experimental section. Density is another significant parameter of NFs that is needed to know the Reynolds number of flowing NFs. Table 3 shows the calculation of the density of NFs. Data shows that the minimal density occurs due to adding a small amount of ZnO NRs to EG.
Table 3. Density of BF (EG), and 0.1, 0.2, and 0.3 vol% ZnO–EG NFs at ambient temperature, 40, 50, 60 °C.
|
Sample Conc.(vol%) |
Ave. weight of BF and NFs at different temperatures, W2 |
Ave. wt. of blank pycnometer (50 mL), W1 |
Density, ρ(g/m3) =(W2-W1)/50 |
|||
|
Ambient Temperature |
40 °C |
50 °C |
60 °C |
|||
|
0 (EG) |
73.30 |
72.99 |
72.77 |
72.76 |
26.67 |
0.932 |
|
0.1 |
73.57 |
73.31 |
73.03 |
72.51 |
26.67 |
0.933 |
|
0.2 |
73.25 |
73.24 |
73.25 |
73.28 |
26.67 |
0.938 |
|
0.3 |
73.85 |
73.52 |
73.27 |
72.83 |
26.67 |
0.944 |
Again, forced convective HTC was measured by varying the temperature of hot water. Figure 7 shows the HTC of 0.1, 0.2, and 0.3 vol% ZnO−EG NFs at different temperatures with standard deviations. If the source of heat (hot water) is kept at higher temperatures, the heat transfer is higher as seen in Figure 7. Average HTC of 2563, 3390, and 3686 were calculated for the hot water temperature of 40 °C, 50 °C, and 60 °C. Therefore, it seems significant to maintain the temperature of heat source fluid at an optimum higher degree.
Figure 7. Forced convective HTC of 0.1, 0.2 and 0.3 vol% ZnO−EG NFs at different temperatures.
Another important factor for heat transfer fluid flow is Reynolds numbers (Re) which need to be considered during NF flow. Figure 8 presents the forced convective HTCs of 0.1, 0.2, and 0.3 vol% ZnO−EG NFs at different Re. Experiments were performed Re at 400, 500, and 600. Data indicate that HTC of the same concentration NFs increases with the increase in Reynolds numbers.
Figure 8. Forced convective HTCs of 0.1, 0.2, and 0.3 vol% ZnO−EG NFs at different Reynolds numbers.
Next, based on the reviewer’s suggestion, we have gone through the literature review again and summarized some papers on using aqueous and non-aqueous (EG and others) based NFs. At a point, the present limitation of using aqueous NFs is mentioned briefly. However, the main aim of our study is not to prescribe the benefit of using EG as base fluid. HTC measurement is the final goal at various concentrations of NFs. As the scope of the present study is limited, in this work, ZnO NPs are synthesized to order to prepare NFs. Then, the stability of the NFs is assessed by some analysis. Finally, the HTC of NFs at different concentrations was calculated, and the conclusion was drawn.
Point 2: Draw the PDF card with XRD results in figure 2.
Response 2: The PDF card with XRD results is added in Figure 2. A new figure is given below:
Figure 2. XRD peak pattern of the products; (a) JCPDS and (b) Synthesized ZnO NRs.
Point 3: In any experiment, uncertainty calculation must be included as a sub-section of
the result section or experimental analysis.
Response 3: In the revised manuscript, brief uncertainty is discussed in the experimental section.
Point 4: Articles use inconsistent units, sometimes millimeters, sometimes centimeters,
sometimes inches. Consistent SI units should be used.
Response 4: In the revised version of the manuscript, uniform and SI units were used.
Point 5: In the experimental test, the setup should be tested and compared with any previous study if available, or the setup is tested with any fluid found in the references to ensure its calibration.
Response 5: A reference was mentioned (Ref. No. 52) which used the same set-up for heat transfer coefficient measurement of CuO-PVA nanofluids and the results are comparable.
Point 6: Thermophysical properties should be explained more and put in the article the
correlation used in calculation.
Response 6: Thermophysical properties are now explained in more detail with new different Figures and Tables.
Point 7: I suggest to add some experiments in the part related to heat transfer, because it is the important part of the paper and there is only one graph (Fig. 6) in it. I suggest that the study be done at different Reynolds numbers (Re) and plot them with the heat transfer coefficient (HTC) and if possible to find the correlation between the Reynolds number (Re) and the Nusselt number (Nu).
Response 7: We added 2 new Figures (Figure 7 & Figure 8) to the HTC section considering the Reynolds number as follows:
Again, forced convective HTC was measured by varying the temperature of hot water. Figure 7 shows the HTC of 0.1, 0.2, and 0.3 vol% ZnO−EG NFs at different temperatures with standard deviations. If the source of heat (hot water) is kept at higher temperatures, the heat transfer is higher as seen in Figure 7. Average HTC of 2563, 3390, and 3686 were calculated for the hot water temperature of 40 °C, 50 °C, and 60 °C. Therefore, it seems significant to maintain the temperature of heat source fluid at an optimum higher degree.
Figure 7. Forced convective HTC of 0.1, 0.2 and 0.3 vol% ZnO−EG NFs at different temperatures.
Another important factor for heat transfer fluid flow is Reynolds numbers (Re) which need to be considered during NF flow. Figure 8 presents the forced convective HTCs of 0.1, 0.2, and 0.3 vol% ZnO−EG NFs at different Re. Experiments were performed Re at 400, 500, and 600. Data indicate that HTC of the same concentration NFs increases with the increase in Reynolds numbers.
Figure 8. Forced convective HTCs of 0.1, 0.2, and 0.3 vol% ZnO−EG NFs at different Reynolds numbers.
However, finding out the relation between Re and Nu is out of the present scope. Nusselt number is not considered here. We hope to do such kind of relation in our future project.
Point 8: The following references are examples of using nanofluids characterization and
heat transfer:
(a) Redhwan Almuzaiqer, Mohamed Elsayed Ali, and Khaled Al‐Salem, “Tilt Angle’s Effects on Free Convection Heat Transfer Coefficient Inside a Water‐Filled Rectangular Parallelepiped Enclosure”, Processes 2022, Volume 10, Issue 2, 396. https://doi.org/10.3390/pr10020396.
(b) Redhwan Almuzaiqer, Mohamed Elsayed Ali, and Khaled Al-Salem “Effect of the Aspect Ratio and Tilt Angle on the Free Con-vection Heat Transfer Coefficient Inside A l2O3–Water-Filled Square Cuboid Enclosures”Nanomaterials 2022, 12(3), 500, pp. 1-32.
https://doi.org/10.3390/nano12030500 . (This article belongs to the Special Issue Impact of Nanofluid on Heat Transfer).
(c) Mohamed Ali, O. Zeitoun and Salem Almotairi, “Natural convection heat transfer inside vertical circular enclosure filled with water-based Al2O3 nanofluids”, Int. Journal of Thermal Sciences , Vol. 63, January 2013, PP 115-124, 2013.
(d) Mohamed Ali, O. Zeitoun, Salem Almotairi and Hany Al-Ansary, “The effect of Alumina-water nanofluid on natural convection heat transfer inside vertical circular enclosure heated from above”, Heat Transfer Engineering, Vol. 34, issue 15, pp. 1289- 1299, 2013.
Response 8: We thank the reviewer again for mentioning the relevant references. We have gone through and added 8 new references (Ref. No. 11, 12, 13, 32, 33, 34, 35, and 40) in our revised manuscript, including the ones mentioned above. We have also rearranged the numbering of references accordingly.
.
Some characterization of NFs is done. For instance, stability assessment (by observation of sedimentation, UV-vis, zeta potential measurement), viscosity, the density of NFs, and heat transfer-related measurements. However, few characterizations mentioned in those papers are not available to us.

Reviewer 4 Report
Very interesting topic. Convective Heat Transfer Coefficient Measurement of ZnO–EG NFs.
Characterization made by Xray, zeta potential and heat flux.
I wonder if you could characterize viscosity, or maybe that is another line of work you can investigate. In diverse devices and components, this is a critical issue which had attracted attention from industry and researchers.
Please work on your english grammar along the manuscript body.
I highly recommend you search for novel references (>2019 - 2020).
Author Response
We gratefully acknowledge the constructive comments and valuable suggestions of the reviewer. We have read the reviewer’s comments carefully and tried our best to revise the manuscript accordingly. The responses to the reviewer’s comments are listed below. In the revised manuscript, the revised/corrected part was highlighted using the “Track Changes” function.
Point 1: Very interesting topic. Convective Heat Transfer Coefficient Measurement of ZnO–EG NFs.
Characterization made by X-ray, zeta potential and heat flux.
I wonder if you could characterize viscosity, or maybe that is another line of work you can investigate. In diverse devices and components, this is a critical issue which had attracted attention from industry and researchers.
Response 1: We would like to thank the reviewer again for his inspiring comments on the topic. We understood the importance of characterizing the viscosity of ZnO−EG NFs. We have measured the viscosity of ZnO–EG NFs and added them to the revised manuscript. Viscosity was measured by a Brookfield Viscometer (DV-II Pro EXTRA) at ambient temperature and at 100 rpm. The results are presented in the manuscript as below:
3.3. Viscosity of ZnO–EG NFs
As shown in Table 2, the average viscosity of BF (EG), and 0.1, 0.2, and 0.3 vol% ZnO–EG NFs at ambient temperature, at 40, 50, and 60 °C, and 100 rpm were found to be 7.40, 7.60, 7.85, and 8.46 cP respectively as shown in Table 2. Relative viscosity was also calculated, which indicates no major differences at different concentrations of NFs. It was found that adding NPs to NFs increased their viscosity by a tiny amount, making them suitable for heat transfer applications with minimal pressure drop in the flow channels. Viscosity values are important to get an idea about the stability of NFs. Also, viscosity data is needed to calculate the Reynolds number of the flow of NFs.
Table 2. The viscosity of BF (EG), and 0.1, 0.2, and 0.3 vol% ZnO–EG NFs at ambient temperature and 100 rpm.
|
Concentration (vol%) |
Viscosity, μ (cP) |
Relative Viscosity, μr |
|
0 (EG) |
7.40 |
- |
|
0.1 |
7.60 |
1.02 |
|
0.2 |
7.85 |
1.06 |
|
0.3 |
8.46 |
1.14 |
Point 2: Please work on your English grammar along the manuscript body.
Response 2: We have checked the English grammar and language again and corrected it to our best.
Point 3: I highly recommend you search for novel references (>2019 - 2020).
.
Response 3: By following the suggestions of the reviewers. We have searched for new references and added relevant 8 references (Ref. No. 11, 12, 13, 32, 33, 34, 35, and 40), including the references from the years mentioned above. We have also rearranged the numbering of references accordingly in the manuscript.
.

Round 2
Reviewer 1 Report
The paper can be accepted in the present form
Reviewer 2 Report
The revision is OK.